# Efficient open recruitment and perspectives of host families on medical student homestays in rural Japan

**Tsuneaki Kenzaka**[1,2]*, **Shinsuke Yahata**[3], **Ken Goda**[1,2], **Ayako Kumabe**[1], **Hozuka Akita**[2], **Masanobu Okayama**[3]

**1** Division of Community Medicine and Career Development, Kobe University Graduate School of Medicine, Kobe, Japan, **2** Department of Internal Medicine, Hyogo Prefectural Tamba Medical Center, Tamba, Japan, **3** Division of Community Medicine and Medical Education, Kobe University Graduate School of Medicine, Kobe, Japan

* smile.kenzaka@jichi.ac.jp

## Abstract

We devised and assessed open recruitment of host families for medical student homestays in a rural area of Hyogo Prefecture, Japan, so that program organizers would not have to depend on professional and personal connections. The duration of the homestays was one night and two days, and they were conducted in August 2016, 2017, and 2018. The purpose of this community-based medical education program was to promote interactions between medical students and residents of Tamba area. The study asked one family member from each host family to complete a questionnaire after the homestay, and their experiences were evaluated in the study. The questionnaire results were analyzed using a visual analog scale (VAS; 0–100 mm). Thirty-three host families participated in the homestay program over three years. Results showed that VAS scores were high for enjoyment of homestays (VAS; 92.4 ± 13.0), continuation of the homestay program (91.7 ± 12.7), continuation of participation in the homestay program (89.2 ± 16.2), and desire for the homestay students to work in the area in the future (95.4 ± 6.3). The recruitment of host families through advertising was an efficient method for this community-based medical education homestay program. The results indicate that it is possible to attract more host families through open recruitment, which will contribute to the sustainability of the homestay program. Further research, including a follow-up of the students who participated and whether they chose a rural area or Tamba to practice is needed in the future. Since this is an ongoing program, further research in a similar format can be conducted in the future.

## Introduction

The Model Core Curriculum for Medical Education in Japan (AY 2016 Revision) describes the importance of community-based medical training [1]. This educational strategy aims "to create opportunities to learn and experience medical care in daily life in the community provided from the perspective of behavioral and social sciences (primarily qualitative) in collaboration

**Data Availability Statement:** All relevant data are within the paper.

**Funding:** The authors received no specific funding for this work.

**Competing interests:** The authors have declared that no competing interests exist.

with anthropology, sociology, psychology, philosophy, and education" [1]. Community-based medical education programs have helped to increase medical students' interest in community medicine [2], as these programs demonstrate the importance of living in an area and interacting with its residents [2,3].

Homestay is one of the most effective means for students to stay in close contact with local residents for a long time. To date, some Japanese medical institutions have implemented community-based medical education programs such as community homestays [4]. Potentially, educational programs that enhance interaction with residents can strengthen the relationships between medical practitioners and their patients in the community. However, to date, there are no data available to support effectiveness of such medical programs [5].

Host families participating in homestay programs in several Japanese communities have been recruited through professional and personal connections with the organizers [4]. Homestay programs allow medical students to experience the lives of their host families, as they learn about the area and its culture [6]. Importantly, studies have revealed that educational homestays influence language learning, cultural immersion, and the development of professional skills of students pursuing careers in health science [5]. Our previous study was the first on community-based medical education programs, which showed that homestays strengthened the relationship between medical students, their host families, and the community [6]. Following were some of the findings of the study: "the program improved the students' attitudes toward practicing community medicine. Moreover, the students appreciated the fact that their training sites could become their workplaces in the future" [6]. However, the study did not consider the perspective of the host families and the method employed for their recruitment. Therefore, there is a need to investigate the homestay program structure, optimal duration, and the best host family recruitment methods. This will help findings with regard to homestay programs to be generalized more effectively.

In August 2016, 2017, and 2018, we conducted a homestay program for medical students in Tamba area (comprising Tamba-sasayama and Tamba city), Hyogo Prefecture, Japan. In community-based medical education, opportunities for medical students to interact with local residents are important [2,3]. By closely interacting with the residents, the attitude of medical students toward community medicine is likely to be enhanced [2,3]. The medical students may also work in the community in future. In theory, an educational program including homestay that promote close interaction with the residents seems to be more effective. Therefore, the medical students stayed in residents' homes for one night and two days. Our results demonstrated that medical students favored community-based medical education programs and that these significantly influenced their attitudes toward community medicine [6]. In addition, the program encouraged candidates to consider working in the homestay area in the future and instilled favorable changes with regard to students' awareness of community medicine [6].

We believe that homestay programs should be generally incorporated into medical students' training, and a methodology for evaluation should be established so that homestay recruitment is sustainable. The present study recruited host families through advertising, and the host families did not have any personal or professional connections with the homestay organizers. Although it may be easier to find host families using personal and professional connections, it may be more difficult for students to benefit from homestays with these families in terms of securing a sufficient number of host families for the participating students, understanding the feelings of local residents who are unaware of the intention of the organizer, and observing their actual lifestyle. For larger numbers of students to be able to participate in homestay programs, public recruitment is necessary, but this process entails some challenges. The present study devised and assessed an efficient method for recruiting host families that does not depend upon the connections of the organizer of the homestay program.

## Materials and methods

### Outline of the educational program

The community-based medical education program (community medicine summer seminar) is offered by the Hyogo Prefectural Tamba Medical Center (referred to as Hyogo Prefectural Kaibara Hospital until July 2019), Tamba, Japan. In addition to the homestay program, the educational program includes off-the-job trainings on abdominal echo and blood sampling technique using a simulator and health lectures for community residents; experiencing local industries, such as catching amago fish and agriculture (experience planting vegetable seedlings); and visiting important local historical sites. The duration of the homestays was one night and two days, and were conducted in August 2016, 2017, and 2018. Tamba city has an area of 493.21 km$^2$ and the population is approximately 60,000; Tamba-sasayama city has an area of 377.59 km$^2$ and the population is approximately 40,000.

In June 2016, 2017, and 2018, we advertised in hospital newsletters and local newspapers to recruit host families. The recruitment period was about one month each year in June (2 months before the homestay program). We then conducted home visits with first-time hosting families, the duration of which was approximately one hour. We provided them with an overview of the program. The homestay organizers verified the amenities with the host families, such as lockable room doors, air conditioning, use of the bathroom, bedding, pets, dinner and breakfast provision, number of host families willing to attend the one-day internship, and pick-up and delivery arrangements for the students. While interviewing the homestay families, one of the authors sought their assurance that the medical students would be allowed to stay with them without any objection or obstacle. We envisioned that 1–2 students could be sent to one family. Since fewer host families were available, we requested all those we interviewed to accept students. The host families volunteering for the second or third time were asked to provide verification of amenities in writing, in lieu of interviews. During the annual open recruitment period, once each year, we directly requested families who had been hosts in the past, to do so once again. Host families gave written consent to host the students.

The participating students were regional quota students who were going to work in Hyogo Prefecture. These students had received scholarships from Hyogo prefecture, during they were medical students for their six years, and were obliged to work in the rural area of Hyogo prefecture for 9 years after becoming doctors, to be exempted from repaying the scholarship. Out of approximately 120 regional quota students, 70–80 of them participated in this community-based medical education program. Their participation in the community-based medical education program was voluntary, and non-participation did not affect their grades or promotion. Each year, out of 70–80 participants, which included students other than regional quota students, 11–15 students were assigned to the Tamba area program. The students (regional or non-regional quota students) were assigned randomly, regardless of their host family or area of Hyogo prefecture preferences. Altogether, there were 39 students assigned to the Tamba area program over the 3-year period, and 38 of them were regional quota students. This is because this community-based medical education program is a good opportunity for regional quota students to learn about the area where they might get the opportunity to work in future. It can also be attributed to the fact that the organizers actively call for participation from regional quota students. Their participation in the community-based medical education program was voluntary; however, regardless of their preference for a homestay program, all students assigned to the Tamba area program were required to partake in the homestay program.

The mean age of the participants, who were students from 1$^{st}$ to 6$^{th}$ years, was 21.2 ± 2.2 years. Factors such as pet and food allergies; kind of host family they wanted to stay with; and host family's preferences such as gender, number of students, and accommodation of pets,

were considered while assigning host families to the students. Host family composition was also considered, as, for instance, a female student could not be accommodated in a house where only males lived. One participant, who was allergic to cats, was placed in a host family without cats.

The host family came to the meeting place of the participants and met the medical students they were to host. The participants went to the houses of the host families in the families' private cars. The medical students stayed overnight at their host families' residence. The next day, the homestay duration was completed, and the students arrived at the meeting place in the host family's private car. he organizer arranged a single car for several host families who did not have a private car for the transportation of the medical students.

During the homestay, the host family had dinner and the next day's breakfast with the medical student and engaged in conversation. In addition, the medical students were instructed to interview their host families about their daily lives, health, and medical problems. On the following day, group discussions were held in Tamba area on the medical issues gleaned through these interviews.

Students were advised to contact the homestay organizers on their mobile phones if an emergency arose. Similarly, the host family was provided emergency contact details in case any student issue arose.

We created such a program to encourage medical students to learn about the culture and history of Tamba area and local residents' expectations from doctors and to raise awareness regarding the role of doctors who contribute to community medicine. After implementing the 2016 program, we confirmed its effectiveness by collecting information from students through questionnaires and by conducting homestay programs in the same way in 2017 and 2018. The only difference was that while recruiting host families, in addition to following the same recruitment procedure, families who had hosted the students in the first year were directly requested to host the students again.

## Study design and participants

In this analytical observational study, the research targets comprised representatives from all host families who agreed to participate in the homestay program. There were 12 host families and 15 medical students in 2016, 11 host families and 12 medical students in 2017, and 10 host families and 11 medical students in 2018. The number of participating students declined over the years because the number of students assigned to areas other than Tamba area increased.

The study was approved by the Ethics Committee of Hyogo Prefectural Kaibara Hospital (approval number: Kai-Byo number 1216). All participants agreed in writing to the presentation of data obtained during the study.

## Measures and procedure

A self-administered questionnaire was provided to each host family before the homestay commenced and one family member was asked to fill in the necessary details. The completed questionnaire was returned by mail or fax within one week of completion of the homestay.

The participants wrote the responses on the questionnaire. Information such as age, sex, whether the respondent required regular medical attention for any illness, whether the respondents or any family members were healthcare professionals, and information about their profession, and major topics of conversation with medical students during the homestay were recorded in the questionnaire.

In addition, participants answered the following seven questions, which were evaluated using a visual analog scale (VAS; 0–100 mm): 1) Did you enjoy hosting the student?; 2) Did

you experience any inconvenience (VAS 0 = yes and VAS 100 = no)?; 3) Do you think the homestay experience was meaningful for students?; 4) Do you think it was meaningful for students to directly interact with Tamba residents?; 5) Do you think the homestay program should be continued?; 6) Will you continue to volunteer for homestays?; and 7) Do you want homestay students to work in this region in future?

Since there were no previous studies found that were relevant, the questions for this study were developed and thoroughly examined by the co-authors before finalizing the contents. The content of the questionnaire was devised with reference to the content of the questionnaire presented to the students who participated [2,6].

### Data analysis

The questions regarding the homestay sought information pertaining to the following: respondent's experience with the homestay (first time or more than once), whether they required regular medical attention for any illness (yes or no), and whether the respondent or any of the family members were healthcare professionals (yes or no). The VAS scores between groups were compared using Mann–Whitney's U test. All analyses were performed using SPSS version 25.0 (IBM, Armonk, NY), and statistical **significance was set at p $<$ 0.05.**

## Results

The numbers of host families were 12, 11, and 10 in 2016, 2017, and 2018, respectively. Questionnaire responses were obtained from representatives of all host families (response rate was 100%). Host families #9 and 5 hosted the students two and three times, respectively, as they responded multiple times over the years. There were no incidents requiring emergency contact during the homestays, nor were there any incidents that led to the discovery of post-homestay issues. The respondents' mean age ± standard deviation (SD) was 62.4 ± 7.8 years. There were 13 males (39.4%); 22 respondents needed regular medical attention for an illness (66.7%); and 23 respondents (69.7%) or family members were healthcare professionals. Details of healthcare professionals are as follows: one nurse, three pharmacists (one served as a host family three times), one dentist, and two radiologists (one whose family served as host family twice) (Table 1).

The main topics of conversation with medical students during homestays are presented in Table 2. Of the respondents, twenty-five (75.8%) responded that discussions focused on students' studies, and 25 (75.8%), 23 (69.7%), and 23 (69.7%) responded that expectations of medical students, expectations of hospitals, and community medicine, respectively, were conversation topics.

**Table 1. Healthcare professions of respondents and family members (individuals may have multiple healthcare practitioner roles, including duplication).**

| Job type | Respondent (# of persons) | Family members (# of persons) | Total (# of persons) |
|---|---|---|---|
| Nurse | 1 | 13 | 14 |
| Physician | 0 | 9 | 9 |
| Pharmacist | 3 | 3 | 6 |
| Dentist | 1 | 3 | 4 |
| Radiologist | 2 | 1 | 3 |
| Technologist | 0 | 2 | 2 |
| Physical therapist | 0 | 2 | 2 |
| Registered dietitian | 0 | 1 | 1 |

**Table 2. Conversation topics with medical students during the homestay.**

| Topics | Number of respondents (%) | Topics | Number of respondents (%) |
|---|---|---|---|
| Expectations of medical students | 25 (75.8) | Social situation | 4 (12.1) |
| Studies | 25 (75.8) | Hobbies | 3 (9.1) |
| Expectations from hospitals | 23 (69.7) | View of life | 3 (9.1) |
| Community medicine | 23 (69.7) | Student life | 2 (6.1) |
| Regional history | 12 (36.4) | Reason for wanting to become a physician | 1 (3.0) |
| Family and personal stories | 12 (36.4) | Aging, geriatric healthcare | 1 (3.0) |
| Local conditions (life and prospects) | 4 (12.1) | Revitalization of region expected from younger generation | 1 (3.0) |
| Health consultation | 4 (12.1) | | |

The responses to the questions about the homestay experience and the VAS scores are shown in Table 3, and high scores were obtained for all questions. The experiences of the host families during the homestays, whether the respondents were regularly being treated for an illness, or whether the respondents or any family members were healthcare professionals were compared between the groups; however, no significant differences were observed in the VAS scores.

## Discussion

We introduced an efficient method for recruiting host families that does not depend on personal or professional connections between homestay families and homestay organizers, and the perceptions of the host families regarding the homestays were gathered and analyzed. This study is the first that has gathered the opinions of residents about a homestay program. The

**Table 3. Homestay experiences and VAS scores.**

| Question | Total (n = 33) | Experiences of the host family — Initial (n = 19) | 2nd or subsequent time (n = 14) | P value | Respondent is a regular outpatient — Yes (n = 21) | No (n = 12) | P value | Respondent or family members are healthcare professionals — Yes (n = 23) | No (n = 10) | P value |
|---|---|---|---|---|---|---|---|---|---|---|
| Did you enjoy hosting the student? | 92.4 ± 13.0 | 94.6 ± 8.5 | 91.9 ± 14.4 | 0.85 | 89.2 ± 15.4 | 97.9 ± 2.7 | 0.26 | 93.3 ± 11.7 | 90.2 ± 15.9 | 0.51 |
| Did you experience any inconvenience? (0 for yes, 100 for no) | 88.8 ± 11.8 | 86.7 ± 14.0 | 90.4 ± 9.1 | 0.63 | 89.3 ± 12.0 | 87.9 ± 11.9 | 0.72 | 88.0 ± 10.8 | 90.7 ± 14.5 | 0.25 |
| Do you think the homestay experience is meaningful for students? | 88.7 ± 10.4 | 90.1 ± 9.2 | 90.4 ± 9.2 | 0.51 | 86.8 ± 11.3 | 92.2 ± 7.8 | 0.24 | 89.8 ± 9.0 | 96.3 ± 13.2 | 0.64 |
| Do you think it is meaningful for students to directly interact with Tamba residents? | 90.8 ± 10.3 | 90.9 ± 9.4 | 92.0 ± 9.7 | 0.77 | 89.3 ± 11.6 | 93.3 ± 7.6 | 0.51 | 92.4 ± 8.6 | 86.8.9 ± 13.6 | 0.32 |
| Do you think the homestay program should be continued? | 91.7 ± 12.7 | 93.1 ± 11.8 | 92.3 ± 10.2 | 0.94 | 88.7 ± 14.9 | 97.1 ± 3.7 | 0.21 | 94.4 ± 8.5 | 85.7 ± 18.4 | 0.18 |
| Will you continue to volunteer for homestays? | 89.2 ± 16.2 | 88.1 ± 18.5 | 90.6 ± 14.4 | 0.58 | 88.5 ± 15.2 | 90.5 ± 18.5 | 0.60 | 89.9 ± 14.9 | 87.7 ± 19.7 | 0.97 |
| Do you want homestay students to work in this region in future? | 95.4 ± 6.3 | 96.7 ± 6.4 | 95.2 ± 5.6 | 0.38 | 94.3 ± 7.2 | 97.3 ± 4.1 | 0.31 | 96.0 ± 6.5 | 94.2 ± 6.1 | 0.28 |

questionnaire results indicated that the homestay host families responded favorably to the program.

The satisfaction survey revealed that the VAS scores correlate well with the discrete 5-point ordinal rating scales [7]. The VAS scores showed a score of above 75 on the homestay experiences survey of the host family, which is considered to correspond to a score of four or higher with a high degree of satisfaction on a discrete 5-point ordinal rating scale for all questions assessing satisfaction. The high VAS scores showed that the host families that volunteered during open recruitment were very satisfied after the homestay. We believe that this recruitment method is effective for host families. Moreover, the homestay program was indicated to be highly effective by the participating students in their questionnaire. Therefore, we evaluated the selection method for the homestay program to be effective.

Hyogo Prefectural Kaibara Hospital, which is located in a rural area of Hyogo Prefecture, has experienced a serious shortage of physicians, with the number of full-time physicians decreasing from 44 in 2004 to 19 in 2009 [8]. Area residents have initiated several campaigns to protect physicians, other healthcare professionals, and hospitals. Their initiatives include establishment of the Society of Pediatric Protection of Hyogo Prefectural Kaibara Hospital, led by mothers with preschool aged children, and the Medical Regeneration Network, led by a group of local medical professionals. These initiatives have been recognized as pioneering movements to protect the hospitals in Japan [8]. The Society of Pediatric Protection of Hyogo Prefectural Kaibara Hospital and Medical Regeneration Network are not directly involved in the recruitment of host families; however, residents of the area not only wish to protect the hospitals and healthcare professionals but also nurture them and the young students.

Even though the host families had no personal connections with the organizers and responded to open calls through advertising, such as hospital newsletters and local newspapers, more than ten host families applied each year. In the pre-homestay interviews, all host families stated that they wanted to help hospitals and wanted medical students who might come to Tamba area as physicians so as to gain awareness of its charms. The mean age ± SD of the respondents was 62.4 ± 7.8 years, and therefore most were close to the end of their child-rearing responsibilities and accepted medical students who were relatively closer to their children's age. The average life expectancy in 2015 in this area was 80.8 years for men and 87.3 years for women, which is almost the same as the Japanese national average. As indicated in responses obtained on the post-homestay questionnaires, host families' opinions of homestays did not change, and the families thought that having a homestay student was fun and they did not face any inconvenience hosting the students. Furthermore, the respondents expressed their desire for the students who stayed with them to work in their region in the future, as confirmed by the high VAS results (95.4 ± 6.3). About 30% of the respondents and their families were not healthcare professionals. It is also important that non-medical host families showed similar results to those related to medical professions. This shows the usefulness of the open recruitment method for seeking support of those host families who do not have any personal connections with the organizer.

Most homestay studies have been conducted outside Japan, in which the period of homestay ranged from three weeks to two years [5]. Studies on homestay that have been conducted earlier pertain to homestays in South Korea, Russia, China, Spain, and the United States of America, and homestays in Tunisia where people go from the United States of America. However, the period of homestay was extremely short in the case of our study, as it is easier for host families to accept short-term homestays, and therefore it is easier to recruit host families.

Regarding the conversation topics that occurred during the homestay, answers of the host families and the students differed, even though they were referring to the same experience. The students' responses [6] included more health consultations (28.2%) than did the host

families' responses (12.1%). However, expectations of medical students (host family—75.8%, students—61.5% [6]) and expectations from hospitals (host family—69.7%, students—51.2% [6]) were higher in the host family. The topics of conversations are subjective, and it is highly likely that the respondents answered based on their memories. Students may have perceived some conversations as referring to health consultations, even if the host family did not intend it. Furthermore, the host family's expectations of medical students and from hospitals may not have been fully communicated to the students, even if the former strongly desire communicating the expectation.

Some studies in Australia and Japan have indicated that short-term educational programs in rural areas are ineffective in shaping career choices and decisions pertaining to internship locations [9], whereas others have reported that such programs are effective [10]. Our community-based medical education program that included homestays in a rural area demonstrated effective results with respect to interest of medical students in selection of their career [6]. This suggests that it is important to understand the needs of residents of a rural area and the experiences that can be gained while working/staying in these areas. How well the residents accept and directly interact with the medical students can have some influence on the students' career decisions. During the homestays, residents shared their expectations with the medical students about the hospitals, and the history and status of medical care in the community.

However, our study also had several limitations. First, many host family members were healthcare professionals. Therefore, it was unclear whether this method can be generalized in areas with few healthcare professionals. However, no significant differences were observed between healthcare and non-healthcare professionals. This result indicates that this process can be implemented even in areas where there are few healthcare professionals. Second, the number of applications for host families was about 10 per year in an area with a population of about 100,000 people. Therefore, it is unclear whether open recruitment will be effective if there is need to recruit a larger number of host families. The background of the Tamba area is also a limitation, as the area had been experiencing insufficient medical care for some time and local residents had a strong desire to protect hospitals and healthcare professionals and to nurture young students and healthcare professionals in the community. In Japanese, the phrase "power of an area" (*chiiki-ryoku* in Japanese) [11] refers to the ability of members of a community, including residents and businesses, to recognize community issues and work with other parties to resolve problems and create value. Securing a place to work, appointing leaders, and creating regions where people can live with peace of mind are especially necessary for enhancing the power of an area [11]. The power of the area in the Tamba region is strong when it comes to medical issues. This is likely to have a positive impact on the effectiveness of the homestay program. However, the power of an area differs from region to region, and it may not apply to homestays in other regions or countries. This is the first report focusing on the recruitment method for host families, but further research, including a follow-up of the students who participated and whether they chose a rural area/Tamba in which to practice, is needed in the future. This program is still ongoing and further research can be conducted in a similar format.

## Conclusion

This study is the first that has gathered the opinions of host families about a homestay program. Following participation in a community-based medical education homestay program, host families that were recruited through advertising completed a self-reported questionnaire. Results indicated that the host families had good experiences hosting the students, did not find their stay in their houses inconvenient, and understood the significance of participation of

medical students in the homestay programs as well as the significance of interacting with the residents. Further, they expressed a strong desire for the homestay medical students to work in their region in future. The host families who gave consent for the open recruitment of the participants had favorable opinions about the homestay even after the homestay ended. Open recruitment through advertising proved to be an efficient method for finding host families. The results of the present study indicate that it is possible to attract more host families through open recruitment. This will allow more medical students to be accepted into homestays and hence ensures sustainability of the homestay program. Further research in a similar format can be conducted, as the homestay program is still ongoing.

## Author Contributions

**Conceptualization:** Tsuneaki Kenzaka, Masanobu Okayama.

**Data curation:** Tsuneaki Kenzaka, Ken Goda, Ayako Kumabe.

**Formal analysis:** Shinsuke Yahata.

**Investigation:** Tsuneaki Kenzaka, Ayako Kumabe.

**Methodology:** Tsuneaki Kenzaka.

**Resources:** Shinsuke Yahata, Ken Goda, Ayako Kumabe.

**Supervision:** Shinsuke Yahata, Hozuka Akita, Masanobu Okayama.

**Writing – original draft:** Tsuneaki Kenzaka.

**Writing – review & editing:** Shinsuke Yahata, Ken Goda, Ayako Kumabe, Hozuka Akita, Masanobu Okayama.

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
