## [Decision Letter · Decision Letter 0]

13 Oct 2021

PONE-D-21-30169Host family perspectives on medical student homestaysPLOS ONE

Dear Dr. Kenzaka,

Thank you for submitting your manuscript to PLOS ONE. After careful consideration, we feel that it has merit but does not fully meet PLOS ONE’s publication criteria as it currently stands. Therefore, we invite you to submit a revised version of the manuscript that addresses the points raised during the review process. More importantly, the reviewers raised questions about clarity of the methods section and other sections as well, please make sure that the presentation of the article is clear. Other important concerns that should be addressed is that any conclusions should be supported by the data and the employed methods. A detailed review of the article is attached by three experts. Please submit your revised manuscript by Nov 27 2021 11:59PM. If you will need more time than this to complete your revisions, please reply to this message or contact the journal office at plosone@plos.org. Please include the following items when submitting your revised manuscript:A rebuttal letter that responds to each point raised by the academic editor and reviewer(s). You should upload this letter as a separate file labeled 'Response to Reviewers'.A marked-up copy of your manuscript that highlights changes made to the original version. You should upload this as a separate file labeled 'Revised Manuscript with Track Changes'.An unmarked version of your revised paper without tracked changes. You should upload this as a separate file labeled 'Manuscript'.

We look forward to receiving your revised manuscript.

Kind regards,

Mohammed Saqr, Ph.D

Academic Editor

PLOS ONE

Journal Requirements:

Reviewers' comments:

Reviewer's Responses to Questions

**Comments to the Author**

1. Is the manuscript technically sound, and do the data support the conclusions?

Reviewer #1: Partly

Reviewer #2: Yes

Reviewer #3: Partly

2. Has the statistical analysis been performed appropriately and rigorously? 

Reviewer #1: Yes

Reviewer #2: Yes

Reviewer #3: Yes

3. Have the authors made all data underlying the findings in their manuscript fully available?

Reviewer #1: Yes

Reviewer #2: Yes

Reviewer #3: No

4. Is the manuscript presented in an intelligible fashion and written in standard English?

Reviewer #1: Yes

Reviewer #2: Yes

Reviewer #3: No

5. Review Comments to the Author

Reviewer #1: The authors described the process of open recruitment for homestay host families to provide opportunities for medical students to learn about community medicine in a rural area. The basis of the study focused on self-reported evaluation of the host families’ experiences. The data were derived from a brief questionnaire from a cohort of 33 participating families. The expectation was to determine the sustainability of this recruitment effort to be part of the medical curriculum at their institution. However, what was unclear was the significance of the program: Why was this program necessary? Recommendation is to provide a brief explanation of how this homestay program will benefit the learning of the medical students while also benefiting the community.

The methodology was explained and may be applicable to other areas who may want to start such a program. The criteria for selection of homestay families included questions about amenities and accommodations. However, how were the homestay families selected? Were there any exclusion criteria? Was there a certain number of candidate homestay families that the authors required for the program, or was this an open recruitment based on how many were interested in participating? How long was the open recruitment for each cohort – was this a rolling recruitment throughout the year or only during certain times of the year? Providing this information would strengthen the methodology component with a framework for other institutions to consider.

From a teaching perspective, aside from interactions with the homestay families (or which the authors noted 7 out of the 33 host families were in healthcare), were clinical experiences offered for the students? What level of students (second year, third year, fourth year medical students) were eligible for this homestay program? If this did not include clinical experiences, please briefly explain why not, especially since the background indicated that the homestay program includes strengthening relationships with the community and encouraging the medical students to practice in their communities.

Overall, based on the date, the responses from participating homestay families were positive. The authors’ statement that this was an “effective” program is not clearly shown by the data. How do the responses translate to “effectiveness”?

Reviewer #2: The manuscript titled “Host family perspectives on medical student homestays” described open recruitment of host families for medical student homestays in Tamba, Japan of one night and two days in August 2016, 2017 and 2018. 100% of host families completed a questionnaire which showed positive responses for enjoyment of homestays, continuation of the homestay program and participation in the program, desire for homestay students to work in the area in the future, and that they thought it was meaningful for the students to interact with Tanba residents.

This is a well-written, interesting manuscript and is a nice complement to the PLOS ONE article by the same authors from the medical student perspective. Homestays can be a great way to introduce students to the rural community, and this manuscript may be of interest to other rural areas, not just in Japan, but perhaps other countries as well. Listed below are suggestions and comments to help clarify some points and to potentially improve the utility of this manuscript.

Title:

• Line 1: Suggest adding “recruitment” and “rural Japan” to the title so it reads: “Host family recruitment and perspectives on medical student homestays in rural Japan” to be an accurate reflection of the manuscript. The method of recruitment was stressed throughout the manuscript.

Abstract:

• Line 29: It states the “results indicate that it is possible to attract more host families through open recruitment than through professional and personal connections with homestay organizers and will enable the sustainability of the homestay program” but they didn’t demonstrate how many host families had been recruited through those channels prior to the open recruitment, so this statement should be removed or revised to better reflect the manuscript. See also conclusion comments.

• Suggest also include positive response to the continuation of the homestay program, if there is space.

Introduction:

• Line 72: It states, “it may be more difficult for students to benefit from homestays with these families (using personal and professional connections)” – please explain this statement more. Is it just the number of host families needed, or were the authors thinking of other difficulties for students to benefit?

Materials and Methods:

• General: Appreciate the detail in the home visits with first-time hosting families and matching process

• Line 82: Would be helpful to know the size of the area of Tamba to get a sense of the density of the population

• Line 94: Please define regional quota students, and what year/level they were. The PLOS ONE article from the medical student perspective mentioned 39 students, 38 of whom were regional quota students. Why was the other student not included in this study? Are the regional quota students a priority and other students invited if there is space?

• Line 95: What proportion of students volunteered to be in the homestay program? Were all students who volunteered for the program able to be placed? Were there more host families than student volunteers, but some students had to double up because of allergies or other issues?

• Lines 112-113: The regional quota students decreased from 15 in 2016 to 12 in 2017 to 11 in 2018. Is this a continuing trend downwards? Are there less regional quota students or are less volunteering for the homestays?

• Line 125: Please state that the survey was composed of the 7 questions listed, if that is correct. If not, please add all questions. Were the host families asked if they thought one night was too short? Were the host families asked if they recruited others for the following year?

Results:

• Line 144: Clarify that host families 9 and 5 filled out the survey multiple times in relation to the number of years they hosted. Did their opinion vary over time?

• Did you try to combine the variables? For example (Respondent is a regular outpatient + Respondent or family members are healthcare professions) vs (Respondent is regular outpatient and respondent or family members are not healthcare professions (this would be interesting)) vs (Respondent is not a regular outpatient + respondent or family members are healthcare professions) vs (Respondent is not a regular outpatient + respondent or family members are not healthcare professions)

Discussion:

• General: when other homestay articles are mentioned – would be helpful to also mention country for context.

• General: Conversation topics were also reported in other paper – was there a difference between the student and host point of view and possible reasons why?

• Perhaps the discussion could emphasize that 30% of respondents and their families had no connection to healthcare

• Line 189: The mean age of the respondents is listed at 62.4 – it would be helpful to know the longevity of residents in the area or in Japan.

• Line 204: This statement is too strong. It would be more accurate to say “demonstrated effective results in medical students’ declared interest in career selection”

• Line 214: add population of the area here to add context of 10 applications per year.

• Line 226: “it may not apply to homestays in other regions” – add countries too - "it may not apply to homestays in other regions or countries"

• Future studies mentioned should include a follow-up of the students who participated and whether they chose a rural area/Tamba to practice

Conclusion:

• Line 236 and see abstract comments: It states the “results indicate that it is possible to attract more host families through open recruitment than through professional and personal connections with homestay organizers and will enable the sustainability of the homestay program” but they didn’t demonstrate how many host families had been recruited through those channels prior to the open recruitment, so this statement should be removed or revised to better reflect the manuscript.

• Would it be possible to add an update on how the program is doing? Is it still ongoing? Expanded?

Tables and Figures:

• Table 2: Why is the first column of topics bolded and the second column not bolded?

• Table 3: The last set of “Respondent or family members are healthcare professionals” – the yes (n-23) and no (n=11) add up to 34, when the total n = 33

Reviewer #3: You deal with a topic that, if effectively addressed in medical school curricula, would help future doctors become better community health practitioners.

Title: The title could be clearer. Do you want to suggest that the impact of recruitment through advertisement is positive?

In the abstract, the objective of the study was “ to promote interactions between medical students and residents of Tanba area” however, in the introduction the authors mentioned that The present study devised and assessed an effective method for recruiting host families that does not depend on homestays organizers’ connections. I am concerned about the validity of the study. The authors need to clarify what exactly they did in the study and what was their rationale and aim?

It seems that the authors have divided the previously published study in PLOSone into the year 2020. They did mention the same in the introduction part “Our previous study was the first on community-based medical education programs and showed that homestays strengthened the relationship between medical students, their host families, and the community [6].” If this is the case then what this study will add to the literature? This section needs more elaboration.

Abstract: this section is clear and concise. It highlights the key findings in the manuscript.

Background: the section sets up the study nicely and is thorough in its literature review.

Methods: I appreciate that the authors used this approach of a community program. They include some details about the first steps in the process, but more information on all of the steps could be helpful. In addition, I would be interested in understanding a little bit more about why they chose the instructional methods that they did. And, was there any ongoing assessment of the effectiveness of this community-based approach, and were modifications made over time?

Details about the instrument used to collect the responses are missing. Who developed the questionnaire, how the items were selected, and how validity was ensured, and so on. Your one objective was assessing the effectiveness of the method of selection but the questionnaire and even the open-ended questionnaire did not address the effectiveness. I want the authors should mention how they assessed the effectiveness of the selection method for homestay.

It would be worth mentioning that participants were undergraduate or postgraduate medical students.

Results: this section is clear and makes appropriate references to the additional material that is available in the tables. I think it would be worth including the qualitative measures for the questionnaire items that you reference.

Discussion: It would help me as a reader to see its described responses and its relation to community healthcare services.

The authors state there are limitations "to the study." It's a small point, but I believe that it would be more precise to say that the limitations are "to our findings" or "to our results."

Tables: these are clear and helpful.

In the Conclusion, it would be useful to put the ideas in the context of what has been previously published in this area. What is novel from this quantitative observational study?

I also suggest careful editing of the manuscript to ensure clarity.

6. PLOS authors have the option to publish the peer review history of their article (what does this mean?). If published, this will include your full peer review and any attached files.

Reviewer #1: No

Reviewer #2: No

Reviewer #3: No

---

## [Author Response · Author response to Decision Letter 0]

23 Nov 2021

Review Comments to the Author

Please use the space provided to explain your answers to the questions above. You may also include additional comments for the author, including concerns about dual publi-cation, research ethics, or publication ethics. (Please upload your review as an attach-ment if it exceeds 20,000 characters)

Reviewer #1: The authors described the process of open recruitment for homestay host families to provide opportunities for medical students to learn about community medi-cine in a rural area. The basis of the study focused on self-reported evaluation of the host families’ experiences. The data were derived from a brief questionnaire from a cohort of 33 participating families. The expectation was to determine the sustainability of this recruitment effort to be part of the medical curriculum at their institution. How-ever, what was unclear was the significance of the program: Why was this program necessary? Recommendation is to provide a brief explanation of how this homestay program will benefit the learning of the medical students while also benefiting the community.

Response:　Thanks for your comment. We have added the significance of this program in the Introduction section.（Page 3, Line 51-52）（Page 4, Line 74-78）

The methodology was explained and may be applicable to other areas who may want to start such a program. The criteria for selection of homestay families included ques-tions about amenities and accommodations. However, how were the homestay families selected? Were there any exclusion criteria? Was there a certain number of candidate homestay families that the authors required for the program, or was this an open re-cruitment based on how many were interested in participating? How long was the open recruitment for each cohort – was this a rolling recruitment throughout the year or only during certain times of the year? Providing this information would strengthen the methodology component with a framework for other institutions to consider.

Response:　Thanks for your valuable comment. We have described the process of se-lection of host families, the required number, recruitment period, etc., in the section “Outline of the educational program.” （Page 5, Line 111-112）（Page 6, Line 119-125）

From a teaching perspective, aside from interactions with the homestay families (or which the authors noted 7 out of the 33 host families were in healthcare), were clinical experiences offered for the students? What level of students (second year, third year, fourth year medical students) were eligible for this homestay program? If this did not include clinical experiences, please briefly explain why not, especially since the back-ground indicated that the homestay program includes strengthening relationships with the community and encouraging the medical students to practice in their communities.

Response:　Thanks for your valuable comments. We revised the contents and have provided details about the educational program in addition to homestay program. We also added details about age and grade of participating students in the section “Outline of the educational program.” （Page 5, Line 103-107）（Page 6-7, Line 127-141）

Overall, based on the date, the responses from participating homestay families were positive. The authors’ statement that this was an “effective” program is not clearly shown by the data. How do the responses translate to “effectiveness”?

Response:　Thanks for your comment. I paraphrased “effective method” to “efficient method” in the summary and text. The program itself was an "effective program" for students, as shown in reference No. 6 in our study. This content is attached as a quota-tion in the main text and left as it is. From the results of the homestay program, we tried not to mention in the text that it was an "effective program". （Title, Line 33, 95, 252, 347）

Reviewer #2: The manuscript titled “Host family perspectives on medical student homestays” described open recruitment of host families for medical student homestays in Tamba, Japan of one night and two days in August 2016, 2017 and 2018. 100% of host families completed a questionnaire which showed positive responses for enjoy-ment of homestays, continuation of the homestay program and participation in the pro-gram, desire for homestay students to work in the area in the future, and that they thought it was meaningful for the students to interact with Tanba residents.

This is a well-written, interesting manuscript and is a nice complement to the PLOS ONE article by the same authors from the medical student perspective. Homestays can be a great way to introduce students to the rural community, and this manuscript may be of interest to other rural areas, not just in Japan, but perhaps other countries as well. Listed below are suggestions and comments to help clarify some points and to poten-tially improve the utility of this manuscript.

Response: Thanks for your review and valuable comments.

Title:

• Line 1: Suggest adding “recruitment” and “rural Japan” to the title so it reads: “Host family recruitment and perspectives on medical student homestays in rural Japan” to be an accurate reflection of the manuscript. The method of recruitment was stressed throughout the manuscript.

Response: Thanks for your comment. We have changed the title as per your suggestion from “Host family perspectives on medical student homestays” to “Efficient open re-cruitment and perspectives of host family on medical student homestays in rural Ja-pan.”

Abstract:

• Line 29: It states the “results indicate that it is possible to attract more host families through open recruitment than through professional and personal connections with homestay organizers and will enable the sustainability of the homestay program” but they didn’t demonstrate how many host families had been recruited through those channels prior to the open recruitment, so this statement should be removed or revised to better reflect the manuscript. See also conclusion comments.

Response:　Thanks for your comment. We added the sentence “Thirty-three host fami-lies participated in the homestay program over three years.” We deleted “than through professional and personal connections with homestay organizers” and revised the statement. （Page 2, Line 29, 36-39）

Suggest also include positive response to the continuation of the homestay program, if there is space.

Response:　Thanks for your comment. We added positive responses to the continuation of the homestay program. （Page 2, Line 36-39）（Page 2, Line 36-39）（Page 2, Line 36-39）（Page 17, Line 333-336）（Page 18, Line 350-351）

Introduction:

• Line 72: It states, “it may be more difficult for students to benefit from homestays with these families (using personal and professional connections)” – please explain this statement more. Is it just the number of host families needed, or were the authors think-ing of other difficulties for students to benefit?

Response:　Thanks for your comment. We added the sentence “in terms of securing sufficient number of host families for the participating students, under-standing the feelings of local residents who are unaware of the intention of the organizer, and ob-serving their actual lifestyle.” （Page 5, Line 91-93）

Materials and Methods:

• General: Appreciate the detail in the home visits with first-time hosting families and matching process 

Response: We have added the details of the process of recruitment of host families and how the program was implemented. （Page 5-7, Line 112-125, 148-153, 162-169）

• Line 82: Would be helpful to know the size of the area of Tamba to get a sense of the density of the population 

 Response: Thanks for your comment. We added details regarding the population and size of the area of Tanba city and Tanba-sasayama city. （Page 5, Line 108-110）

• Line 94: Please define regional quota students, and what year/level they were. The PLOS ONE article from the medical student perspective mentioned 39 students, 38 of whom were regional quota students. Why was the other student not included in this study? Are the regional quota students a priority and other students invited if there is space? 

Response: Thanks for your comment. We have added details of the regional quota stu-dents. We have also added details regarding the year of study of the medical students. We added the reason 38 regional quota students of 39 students. （Page 6-7, Line 126-141）

• Line 95: What proportion of students volunteered to be in the homestay program? Were all students who volunteered for the program able to be placed? Were there more host families than student volunteers, but some students had to double up because of allergies or other issues?

Response: Thanks for your comment. We have added details about the regional quota students, participants, about the community-based medical education program as a whole and about Tamba's homestay program. All the students assigned to the Tamba area program took part in the homestay program without any particular desire to partic-ipate in homestay program. Only one of the participants was allergic to cats and we considered assigning them to host families who do not have cats. （Page 6-7, Line 126-140, 146-147）

• Lines 112-113: The regional quota students decreased from 15 in 2016 to 12 in 2017 to 11 in 2018. Is this a continuing trend downwards? Are there less regional quota stu-dents or are less volunteering for the homestays?

Response:　Thanks for your comment. I have added details to present the reason be-hind such a trend. Please find the reason below.

“The number of participating students declined over the years, because the number of students assigned to areas other than Tamba area increased.” （Page 8, Line 175-177）

• Line 125: Please state that the survey was composed of the 7 questions listed, if that is correct. If not, please add all questions. Were the host families asked if they thought one night was too short? 

Response:　Thanks for your comment. Yes, the survey was composed of the 7 ques-tions listed. We did not ask this question.

Were the host families asked if they recruited others for the following year?

Response:　Thanks for your comment. We did not ask this question. During the annual open recruitment period, we asked if the previous host families could rejoin. （Page 6, Line 122-125）

Results:

• Line 144: Clarify that host families 9 and 5 filled out the survey multiple times in re-lation to the number of years they hosted. Did their opinion vary over time?

Response:　Thanks for your comment. We added the sentence “They have responded multiple times in relation to the number of years.” （Page 10, Line 214-215）

2nd or subsequent time in Table 3 summarizes the results of the 2nd -9 people and the 3rd- 5 people. Throughout the 2nd and 3rd times, the answers did not change signifi-cantly over time. Therefore, we presented the sum of the second and third rounds in Table 3.

• Did you try to combine the variables? For example (Respondent is a regular outpa-tient + Respondent or family members are healthcare professions) vs (Respondent is regular outpatient and respondent or family members are not healthcare professions (this would be interesting)) vs (Respondent is not a regular outpatient + respondent or family members are healthcare professions) vs (Respondent is not a regular outpatient + respondent or family members are not healthcare professions)

Response:　Thanks for your comment. We tried to combine the variables. However, all VAS values were high, so, no remarkable difference was observed because of the com-bination.

Discussion:

• General: when other homestay articles are mentioned – would be helpful to also men-tion country for context.

Response:　Thanks for your comment. We added the countries for context. （Page 15-16, Line 298-301, 304）

• General: Conversation topics were also reported in other paper – was there a differ-ence between the student and host point of view and possible reasons why?

Response: Thanks for your comment. We were not able to find the topics of conversa-tion in other homestay articles.

• Perhaps the discussion could emphasize that 30% of respondents and their families had no connection to healthcare. 

Response: Thanks for your comment. We added the sentence “About 30% of the re-spondents and their families were not healthcare professionals. It is also important that non-medical host families showed similar results to those related to medical pro-fessions. This shows the usefulness of the open recruitment method for seeking support of those host families who do not have any personal connections with the organizer.”

（Page 15, Line 292-296）

• Line 189: The mean age of the respondents is listed at 62.4 – it would be helpful to know the longevity of residents in the area or in Japan.

Response: Thanks for your comment. We added the average life expectancy details of residents of the area. （Page 15, Line 286-287）

• Line 204: This statement is too strong. It would be more accurate to say “demonstrat-ed effective results in medical students’ declared interest in career selection”

Response:　Thanks for your comment. We paraphrased as “demonstrated effective re-sults with respect to interest of medical students in selection of their career”（Page 16, Line 307-308）

• Line 214: add population of the area here to add context of 10 applications per year.

Response: Thanks for your comment. We added the population of the area. （Page 16, Line 319-320）

• Line 226: “it may not apply to homestays in other regions” – add countries too - "it may not apply to homestays in other regions or countries" 

Response: Thanks for your comment. We added “or countries.” （Page 17, Line 332）

• Future studies mentioned should include a follow-up of the students who participated and whether they chose a rural area/Tamba to practice

Response:　Thanks for your comment. We added the sentence “further research includ-ing a follow-up of the students who participated and whether they chose a rural ar-ea/Tamba to practice is needed in the future.” （Page 17, Line 333-335）

Conclusion:

• Line 236 and see abstract comments: It states the “results indicate that it is possible to attract more host families through open recruitment than through professional and per-sonal connections with homestay organizers and will enable the sustainability of the homestay program” but they didn’t demonstrate how many host families had been re-cruited through those channels prior to the open recruitment, so this statement should be removed or revised to better reflect the manuscript.

Response: Thanks for your comment. We deleted the sentence “than through personal and professional connections with organizers”.

We left following sentence; “The results of the present study indicate that it is possible to attract more host families through open recruitment.” （Page 17, Line 348-349）

• Would it be possible to add an update on how the program is doing? Is it still ongo-ing? Expanded?

Response: Thanks for your comment. This program is still ongoing and hence further research can be conducted in a similar format. We added this in the Conclusion section and while stating the limitations. （Page 17-18, Line 335-336, 350-351 ）

Tables and Figures:

• Table 2: Why is the first column of topics bolded and the second column not bolded?

Response: Thanks for your comment. We have made the necessary changes. (Table 2)

• Table 3: The last set of “Respondent or family members are healthcare professionals” – the yes (n-23) and no (n=11) add up to 34, when the total n = 33

Response: Thanks for your comment. We made the necessary correction which is as follows: the yes (n=23) and no (n=10) add up to 33. no (n=10) (Table 3)

Reviewer #3: You deal with a topic that, if effectively addressed in medical school cur-ricula, would help future doctors become better community health practitioners.

Response:　Thanks for your valuable comment.

Title: The title could be clearer. Do you want to suggest that the impact of recruitment through advertisement is positive? 

Response:　Thanks for your comment. We changed the title from “Host family per-spectives on medical student homestays” to “Efficient open recruitment and perspec-tives of host family on medical student homestays in rural Japan” to emphasize the im-pact of recruitment.

In the abstract, the objective of the study was “ to promote interactions between medi-cal students and residents of Tanba area” however, in the introduction the authors men-tioned that The present study devised and assessed an effective method for recruiting host families that does not depend on homestays organizers’ connections. I am con-cerned about the validity of the study. The authors need to clarify what exactly they did in the study and what was their rationale and aim?

Response:　Thanks for your comment. We changed "We introduced" to "We devised and assessed" at the beginning of the summary. In addition, we clearly stated that the purpose was "The purpose of community-based medical education program" rather than the purpose of research. （Page 2, Line 21, 24-25）

It seems that the authors have divided the previously published study in PLOSone into the year 2020. They did mention the same in the introduction part “Our previous study was the first on community-based medical education programs and showed that homestays strengthened the relationship between medical students, their host families, and the community [6].” If this is the case then what this study will add to the litera-ture? This section needs more elaboration.

Response: Thanks for your comment. We mentioned the gist of the previous study as following; “the program improved the students’ attitudes toward practicing community medicine. Moreover, the students appreciated the fact that their training sites could be-come their workplaces in the future” [6]. （Page 3-4, Line 21, 65-67）

We also added content about host families that was not sufficiently discussed in the previous study. （Page 4, Line 67-71）

Abstract: this section is clear and concise. It highlights the key findings in the manu-script.

Response: Thanks for your comment.

Background: the section sets up the study nicely and is thorough in its literature re-view.

Response: Thanks for your comment.

Methods: I appreciate that the authors used this approach of a community program. They include some details about the first steps in the process, but more information on all of the steps could be helpful. In addition, I would be interested in understanding a little bit more about why they chose the instructional methods that they did. And, was there any ongoing assessment of the effectiveness of this community-based approach, and were modifications made over time?

Details about the instrument used to collect the responses are missing. Who developed the questionnaire, how the items were selected, and how validity was ensured, and so on. Your one objective was assessing the effectiveness of the method of selection but the questionnaire and even the open-ended questionnaire did not address the effective-ness. I want the authors should mention how they assessed the effectiveness of the se-lection method for homestay.

Response: Thanks for your comment. We have added detailed information about homestay implementation in the section “Outline of the educational program”. （Page 5-7, Line 101-107, 111-113, 119-125, 134-141, 148-153）

 In the introduction section, we added the meaning of homestay as following "Homestay is one of the most effective means for students to stay in close contact with local residents for a long time." （Page 3, Line 51-52）

We created such a program to encourage medical students to learn about the culture and history of Tamba area and local residents’ expectations from doctors, and to raise awareness regarding the role of doctors who contribute to community medicine. （Page 7-8, Line 162-169）

We continued to evaluate the effectiveness of this community-based approach, and in addition to the regular open recruitment of host families, we request a host family di-rectly from a family who has experienced a host family once. （Page 6, Line 123-125）

Other than that, we stated that the method of recruitment was the same every year.

Since there were no previous studies found which were relevant, the questions for this study were developed and thoroughly examined by the co-authors before finalizing the contents. The content of the questionnaire was devised with reference to the content of the questionnaire pre-sented to the students who participated [2,6]. （Page 9, Line 198-201）

The satisfaction survey revealed that the VAS scores correlate well with the discrete 5-point ordinal rating scales [7]. The VAS scores showed an s score of above 75 on the homestay expe-riences survey of the host family, which is considered to correspond to a score of 4 or higher with a high degree of satisfaction on a discrete 5-point ordinal rating scale for all questions as-sessing satisfaction. The high VAS scores showed that the host families that volunteered during open recruitment were very satisfied after the homestay. We believe that this recruitment meth-od is effective for host families. Moreover, the homestay program was indicated to be highly effective by the participat-ing students in their questionnaire. Therefore, we evaluated the selection method for the homestay program to be effective. (Page 14, Line 258-266）

It would be worth mentioning that participants were undergraduate or postgraduate medical students.

Response:　Thanks for your comment. We presented the grades and ages of the partic-ipating medical students in the section “Outline of the educational program.” (Page 7, Line 140-141）

Results: this section is clear and makes appropriate references to the additional materi-al that is available in the tables. I think it would be worth including the qualitative measures for the questionnaire items that you reference.

Response:　Thanks for your comment. 

Discussion: It would help me as a reader to see its described responses and its relation to community healthcare services.

The authors state there are limitations "to the study." It's a small point, but I believe that it would be more precise to say that the limitations are "to our findings" or "to our results."

Response:　Thanks for your valuable comment. We　rephrased to “However, our study also had several limitations” (Page 16, Line 314）

Tables: these are clear and helpful.

Response: Thanks for your comment.

In the Conclusion, it would be useful to put the ideas in the context of what has been previously published in this area. What is novel from this quantitative observational study?

Response: Thanks for your comment. This study is the first that has gathered the opin-ions of host family about a homestay program. Also, the host families who participated in the open recruitment had favorable opinions about the homestay even after the end of the homestay. These findings are novel. (Page 17, Line 338-339, 345-346）

I also suggest careful editing of the manuscript to ensure clarity.

Response: Thanks for your comment. We sought assistance of a native English editor to proofread the revised manuscript

---

## [Decision Letter · Decision Letter 1]

17 Dec 2021

PONE-D-21-30169R1Efficient open recruitment and perspectives of host family on medical student homestays in rural JapanPLOS ONE

Dear Dr. Kenzaka,

Thank you for submitting your manuscript to PLOS ONE. After careful consideration, we feel that it has merit but does not fully meet PLOS ONE’s publication criteria as it currently stands. Therefore, we invite you to submit a revised version of the manuscript that addresses the points raised during the review process. While reviewers have recognised the efforts made to improve the manuscripts, they raised some minor comments and suggestions that would improve the article. Addressing these comments in a satisfactory way would help me make a positive decision in a timely way. 

We look forward to receiving your revised manuscript.

Kind regards,

Mohammed Saqr, Ph.D

Academic Editor

PLOS ONE

Journal Requirements:

Reviewers' comments:

Reviewer's Responses to Questions

**Comments to the Author**

1. If the authors have adequately addressed your comments raised in a previous round of review and you feel that this manuscript is now acceptable for publication, you may indicate that here to bypass the “Comments to the Author” section, enter your conflict of interest statement in the “Confidential to Editor” section, and submit your "Accept" recommendation.

Reviewer #1: All comments have been addressed

Reviewer #2: (No Response)

Reviewer #3: All comments have been addressed

2. Is the manuscript technically sound, and do the data support the conclusions?

Reviewer #1: Yes

Reviewer #2: Yes

Reviewer #3: Yes

3. Has the statistical analysis been performed appropriately and rigorously? 

Reviewer #1: Yes

Reviewer #2: Yes

Reviewer #3: Yes

4. Have the authors made all data underlying the findings in their manuscript fully available?

Reviewer #1: Yes

Reviewer #2: Yes

Reviewer #3: Yes

5. Is the manuscript presented in an intelligible fashion and written in standard English?

Reviewer #1: Yes

Reviewer #2: Yes

Reviewer #3: Yes

6. Review Comments to the Author

Reviewer #1: The revised manuscript is much clearer and stronger with the revision made, including describing the students who participated in the program (pages 18-19, lines 126-147), expectation of students to interview their host families about their daily lives and health, and group discussions about the medical issues (page 19, lines 155-158). With the statistics of fewer physicians in the rural areas, it would be interesting to see in future studies by the authors about how many (if any) of the student participants in this program (1) volunteer to participate again but in another rural area and/or (2) continue to serve in these communities that they experienced the homestay after they graduate.

Please see below some additional minor recommended suggestions.

Comments on outline of program:

(1) Page 15, line 52 – “Till date, some…” may be clearer if written as ‘To date, some Japanese medical…”; same suggestion for line 55 to use “to date…” instead of “till date…”

(2) Page 18, line 19 – rather than “I,” suggest using “one of the authors” or “a senior author” sought their assurance in order to maintain consistency in third-person grammar usage

(3) Page 18, lines 134-135 – please clarify, “The students were assigned regardless of their wishes,…” to what is this regarding – which host family to stay with or which area of Hyogo prefecture or some other request by students? Or, is this phrase referring to whether a student is a regional quota student or a non-regional quota student, suggesting that all students have the opportunity to participate?

(4) Page 19, lines 139-140 – if the program was voluntary, please explain the sentence about students in the Tamba area program who “took part in the homestay program without any particular desire to participate in it.” This seems contradictory that “participation in the community-based medical education program was voluntary…” (page 18, lines 1310132)

(5) Page 19, line 150 – the sentence about bathing and sleeping at the homestay can be revised to state that the students stayed overnight at their host families’ residence.

Overall body of manuscript – please review one more time the sentence structure such as extra periods, etc.

Reviewer #2: Thank you for the revisions of this manuscript. Listed below are requests for clarification, grammatical suggestions and comments.

Title:

• Line 1: I believe family should be pleural – “Efficient open recruitment and perspectives of host families on medical student homestays in rural Japan”

Introduction:

• Lines 52 and 55-56: It should be “To date” instead of “Till date”

Materials and Methods:

• Lines 123-125: Please clarify the sentence, “During the annual open recruitment period, we request a host family directly from a family who has experienced a host family once”. Do you mean that you ask families who have experienced being a host family in the past? Or do you mean that you ask families whose children were students who experienced a host family? A student would experience a host family, while the family would experience being a host family.

• Lines 127-128: Please clarify the sentence, “These students were entitled to study funds from Hyogo prefecture after six years of their enrollment, and were obliged to work in the rural area of Hyogo prefecture for 9 years after becoming a doctor in order to be exempted from repaying the funds”. Part of my confusion may be the different structure of medical school, but I believe that medical school is 6 years long in Japan. It sounds like students received study funds after they were in school for six years – so they paid tuition each year and when they graduated, then they received study funds – amount not specified. What are study funds used for? For the nine years of service, does residency count/is there a residency program, or is this after residency?

• Lines 134-135: Perhaps instead of saying students were assigned “regardless of their wishes”, it could say students were assigned randomly?

• Lines 139-140: Perhaps instead of saying “without any particular desire to participate in it”, it could say “All the students assigned to the Tamba area program were required to take part in the homestay program”.

• Lines 146-147: when you use the word “considered” it gives the impression that you thought about it but did it anyway. If the student who was allergic to cats was not placed in a family with cats, it could say, “Only one of the participants was allergic to cats and they were placed in a host family without cats.”

• Line 189: There is a period after the word “collected” that should be removed

Discussion:

• Line 299: A space appears to be missing between “…Spain, and” and “the United…”

• Line 315: Should be plural - “healthcare professionals”

• Reviewer #2, Discussion, second bullet point: The other paper referred to by this reviewer was the article published by the same authors about students (reference 6 – Kenzaka et al, 2020). There were conversations topics reported in this article from the student perspective – was there a difference between the student and host point of view, and if so, possible reasons why?

Reviewer #3: I have reviewed the article. The authors have addressed all the points and it is worth to publish. This is a very important topic and good way to implement community based teaching.

7. PLOS authors have the option to publish the peer review history of their article (what does this mean?). If published, this will include your full peer review and any attached files.

Reviewer #1: No

Reviewer #2: No

Reviewer #3: No

---

## [Author Response · Author response to Decision Letter 1]

7 Jan 2022

Reviewer #1: The revised manuscript is much clearer and stronger with the revision made, including describing the students who participated in the program (pages 18-19, lines 126-147), expectation of students to interview their host families about their daily lives and health, and group discussions about the medical issues (page 19, lines 155-158). With the statistics of fewer physicians in the rural areas, it would be interesting to see in future studies by the authors about how many (if any) of the student participants in this program (1) volunteer to participate again but in another rural area and/or (2) continue to serve in these communities that they experienced the homestay after they graduate.

Response: We are very grateful for your valuable peer review and for the mention of future studies. The following comments have also been corrected.

Please see below some additional minor recommended suggestions.

Comments on outline of program:

(1) Page 15, line 52 – “Till date, some…” may be clearer if written as ‘To date, some Japanese medical…”; same suggestion for line 55 to use “to date…” instead of “till date…”

Response: Thank you for your comment. We have changed “till date” to “to date” in the text (Page 3, Lines 50, 53–54).

(2) Page 18, line 119 – rather than “I,” suggest using “one of the authors” or “a senior author” sought their assurance in order to maintain consistency in third-person grammar usage

Response: Thank you for your comment. We have changed “I” to “one of the authors” (Page 6, Line 117).

(3) Page 18, lines 134-135 – please clarify, “The students were assigned regardless of their wishes,…” to what is this regarding – which host family to stay with or which area of Hyogo prefecture or some other request by students? Or, is this phrase referring to whether a student is a regional quota student or a non-regional quota student, suggesting that all students have the opportunity to participate?

Response: Thank you for your comment. We have changed this to “The students (regional quota students or non-regional quota students) were assigned randomly and, regardless of their wishes about which host family to stay with or which area of Hyogo prefecture preferences.” (Page 6, Lines 133–135).

(4) Page 19, lines 139-140 – if the program was voluntary, please explain the sentence about students in the Tamba area program who “took part in the homestay program without any particular desire to participate in it.” This seems contradictory that “participation in the community-based medical education program was voluntary…” (page 18, lines 131-132)

Response: Thank you for your comment. This has accordingly been changed to “Their participation in the community-based medical education program was voluntary; however, regardless of their preference for a homestay program, all students assigned to the Tamba area program were required to partake in the homestay program.” (Page 7, Lines 139–142).

(5) Page 19, line 150 – the sentence about bathing and sleeping at the homestay can be revised to state that the students stayed overnight at their host families’ residence.

Response: Thank you for your comment. We have changed this to “The medical students stayed overnight at their host families’ residence.” (Page 7, Line 152).

Overall body of manuscript – please review one more time the sentence structure such as extra periods, etc.

Response: Thank you for your comment. 

We have removed an unnecessary period (Page 9, Line 193).

We added a space between “…Spain, and” and “the United…” (Page 15, Line 297).

We pluralized “healthcare professionals” (Page 17, Line 325). 

We reviewed the text again and hired a professional editing company to ensure the language is professional.

Reviewer #2: Thank you for the revisions of this manuscript. Listed below are requests for clarification, grammatical suggestions and comments.

Title:

• Line 1: I believe family should be pleural – “Efficient open recruitment and perspectives of host families on medical student homestays in rural Japan”

Response: Thank you for your comment. We have changed this to “Efficient open recruitment and perspectives of host families on medical student homestays in rural Japan” (Page 1, Line 1).

Introduction:

• Lines 52 and 55-56: It should be “To date” instead of “Till date”

Response: Thank you for your comment. We have changed “till date” to “to date” in the text (Page 3, Lines 50, 53–54).

Materials and Methods:

• Lines 123-125: Please clarify the sentence, “During the annual open recruitment period, we request a host family directly from a family who has experienced a host family once”. Do you mean that you ask families who have experienced being a host family in the past? Or do you mean that you ask families whose children were students who experienced a host family? A student would experience a host family, while the family would experience being a host family.

Response: Thank you for your comment. We have changed this to “once each year, we directly requested families who had been hosts in the past, to do so once again.” (Page 6, Lines 122–123).

• Lines 127-128: Please clarify the sentence, “These students were entitled to study funds from Hyogo prefecture after six years of their enrollment, and were obliged to work in the rural area of Hyogo prefecture for 9 years after becoming a doctor in order to be exempted from repaying the funds”. Part of my confusion may be the different structure of medical school, but I believe that medical school is 6 years long in Japan. It sounds like students received study funds after they were in school for six years – so they paid tuition each year and when they graduated, then they received study funds – amount not specified. What are study funds used for? For the nine years of service, does residency count/is there a residency program, or is this after residency?

Response: Thank you for your comment. We changed this to “These students had received scholarships from Hyogo prefecture, were medical students in their sixth year enrollment, and were obliged to work in the rural area of Hyogo prefecture for 9 years after becoming doctors, to be exempted from repaying the scholarship.” (Page 6, Lines 126–128).

• Lines 134-135: Perhaps instead of saying students were assigned “regardless of their wishes”, it could say students were assigned randomly?

Response: Thank you for your comment. We have changed this to “The students (regional or non-regional quota students) were assigned randomly, regardless of their host family or area of Hyogo prefecture preferences” (Page 6, Lines 133–135). 

• Lines 139-140: Perhaps instead of saying “without any particular desire to participate in it”, it could say “All the students assigned to the Tamba area program were required to take part in the homestay program”.

Response: Thank you for your comment. We have changed this to “Their participation in the community-based medical education program was voluntary; however, regardless of their preference for a homestay program, all students assigned to the Tamba area program were required to partake in the homestay program” (Page 7, Lines 139–142). 

• Lines 146-147: when you use the word “considered” it gives the impression that you thought about it but did it anyway. If the student who was allergic to cats was not placed in a family with cats, it could say, “Only one of the participants was allergic to cats and they were placed in a host family without cats.”

Response: Thank you for your comment. We have changed this to “One participant, who was allergic to cats, was placed in a host family without cats.” (Page 7, Lines 148–149).

• Line 189: There is a period after the word “collected” that should be removed

Response: Thank you for your comment. We have removed the period (Page 9, Line 193).

Discussion:

• Line 299: A space appears to be missing between “…Spain, and” and “the United…”

Response: Thank you for the comment. We have added a space between “…Spain, and” and “the United…” (Page 15, Line 297).

• Line 315: Should be plural - “healthcare professionals”

Response: Thank you for your comment. We have pluralized this to “healthcare professionals” (Page 17, Line 325). 

• Reviewer #2, Discussion, second bullet point: The other paper referred to by this reviewer was the article published by the same authors about students (reference 6 – Kenzaka et al, 2020). There were conversations topics reported in this article from the student perspective – was there a difference between the student and host point of view, and if so, possible reasons why?

Response: Thank you for your comment. We have added the following: Regarding the conversation topics that occurred during the homestay, answers of the host families and the students differed, even though they were referring to the same experience. The students’ responses [6] included more health consultations (28.2%) than did the host families’ responses (12.1%). However, expectations of medical students (host family—75.8%, students—61.5% [6]) and expectations from hospitals (host family—69.7%, students—51.2% [6]) were higher in the host family. The topics of conversations are subjective, and it is highly likely that the respondents answered based on their memories. Students may have perceived some conversations as referring to health consultations, even if the host family did not intend it. Furthermore, the host family’s expectations of medical students and from hospitals may not have been fully communicated to the students, even if the former strongly desire communicating the expectation.” (Page 16, Lines 302–312).

Reviewer #3: I have reviewed the article. The authors have addressed all the points and it is worth to publish. This is a very important topic and good way to implement community based teaching.

Response: We are very grateful for your valuable peer review.

---

## [Editor Report · Decision Letter 2]

13 Jan 2022

Efficient open recruitment and perspectives of host families on medical student homestays in rural Japan

PONE-D-21-30169R2

Dear Dr. Kenzaka,

We’re pleased to inform you that your manuscript has been judged scientifically suitable for publication and will be formally accepted for publication once it meets all outstanding technical requirements.

Kind regards,

Mohammed Saqr, Ph.D

Academic Editor

PLOS ONE
---

## [Editor Report · Acceptance letter]

17 Jan 2022

PONE-D-21-30169R2 

Efficient open recruitment and perspectives of host families on medical student homestays in rural Japan 

Dear Dr. Kenzaka:

I'm pleased to inform you that your manuscript has been deemed suitable for publication in PLOS ONE. Congratulations! Your manuscript is now with our production department. 

Kind regards, 

on behalf of

Dr. Mohammed Saqr 

Academic Editor

PLOS ONE